# The exquisite specificity of human protein arginine methyltransferase 7 (PRMT7) toward Arg-X-Arg sites

**Timothy J. Bondoc**[ORCID]**, Troy L. Lowe**[ORCID]**, Steven G. Clarke**[ORCID]*

Department of Chemistry and Biochemistry and the Molecular Biology Institute, University of California Los Angeles, Los Angeles, California, United States of America

* clarke@chem.ucla.edu

**Data Availability Statement:** All relevant data are within the paper and its Supporting information files.

**Funding:** This work was supported by the National Science Foundation grant MCB-1714569 (to S. G.

## Abstract

Mammalian protein arginine methyltransferase 7 (PRMT7) has been shown to target substrates with motifs containing two arginine residues separated by one other residue (RXR motifs). In particular, the repression domain of human histone H2B (29-RKRSR-33) has been a key substrate in determining PRMT7 activity. We show that incubating human PRMT7 and [$^{3}$H]-AdoMet with full-length *Xenopus laevis* histone H2B, containing the substitutions K30R and R31K (RKRSR to RRKSR), results in greatly reduced methylation activity. Using synthetic peptides, we have now focused on the enzymology behind this specificity. We show for the human and Xenopus peptide sequences 23–37 the difference in activity results from changes in the $V_{max}$ rather than the apparent binding affinity of the enzyme for the substrates. We then characterized six additional peptides containing a single arginine or a pair of arginine residues flanked by glycine and lysine residues. We have corroborated previous findings that peptides with an RXR motif have much higher activity than peptides that contain only one Arg residue. We show that these peptides have similar apparent $k_m$ values but significant differences in their $V_{max}$ values. Finally, we have examined the effect of ionic strength on these peptides. We found the inclusion of salt had little effect on the $V_{max}$ value but a considerable increase in the apparent $k_m$ value, suggesting that the inhibitory effect of ionic strength on PRMT7 activity occurs largely by decreasing apparent substrate-enzyme binding affinity. In summary, we find that even subtle substitutions in the RXR recognition motif can dramatically affect PRMT7 catalysis.

## Introduction

Post-translational modifications (PTMs) of proteins are chemical modifications ranging from single methyl groups to entire proteins (i.e., ubiquitin) that occur after a protein is translated from messenger RNA [1]. These modifications expand the chemical properties of the twenty canonical amino acids encoded by the DNA, further diversifying the structure and function of the proteome [1]. Protein arginine methylation, for example, is a widespread post-translational

C.) and by funds from the UCLA Academic Senate Faculty Research Program, the Life Extension Foundation, Inc, and the Elizabeth and Thomas Plott Chair in Gerontology of the UCLA Longevity Center (to S. G. C.). T. L was supported by the National Institutes of Health Ruth L. Kirschstein National Research Service Award GM007185. T.B. was supported by funds from the Lorraine and Masuo Toji and Boyer Summer Research Fellowships of the UCLA Department of Chemistry and Biochemistry. The funders had no role in study design, data collection and analysis, decision to publish, or preparation of the manuscript.

**Competing interests:** The authors have declared that no competing interests exist.

modification in mammalian cells with suggested regulatory roles in cellular functions such as transcription, translation, cell signaling, DNA repair and protein stability [2].

Protein arginine methyltransferases (PRMTs) modify peptidyl arginine residues by transferring methyl groups from $S$-adenosyl-$L$-methionine onto the ω-nitrogen atoms of the guanidino group of the arginine side chain [3, 4]. To date, nine PRMTs have been identified in mammals [5]. The PRMTs are classified as type I (PRMTs 1–4, 6, 8), type II (PRMTs 5 and 9), or type III (PRMT7) based on whether they form asymmetric dimethylarginine (ADMA), symmetric dimethylarginine (SDMA), or ω-monomethylarginine (MMA), respectively [6]. Protein arginine methylation proceeds through MMA as an intermediate, after which a second methylation reaction occurs to form either ADMA or SDMA [7]. As a unique member of the PRMT family, PRMT7 is the only known type III PRMT and forms only MMA as product [8].

Recent work has demonstrated that PRMT7 is optimally active under non-physiological temperature, pH, and ionic strength conditions [9–11]. However, more insight is needed to understand PRMT7's broader biological role. PRMT7 has been implicated in cancer metastasis and in particular with certain forms of breast cancer [12–15]. More recently, one group reported that PRMT7 is likely involved in maintaining leukemia stem cells in chronic myeloid leukemia [16]. Finally, evidence has been presented that PRMT7 activity can prime the activity of PRMT5 [17].

Recently, there has been much interest in identifying the physiological substrates of PRMT7. For instance, studies of PRMT7 activity in human cell lines including MCF7, HEK293T, and HeLa proposed that the cytosolic proteins eIF2α and HSP70 are probable *in vivo* substrates, suggesting that PRMT7 occupies some regulatory role in cellular stress response [18, 19]. However, multiple *in vitro* studies also suggest that core histones are good substrates of PRMT7, of which histones H2B and H4 are particularly good methyl acceptors [8, 10, 20]. Feng and coworkers further identified the sites in histone H2B that PRMT7 recognizes as the lysine- and arginine-rich regions containing an Arg-X-Arg motif (RXR), consisting of two Arg residues separated by any other amino acid, which exists near the N-terminal tail of histone H2B with the sequence 29-RKRSR-33 [8, 11]. Thus, understanding the specificity of PRMT7 toward basic RXR-containing regions could be helpful in identifying additional *in vivo* substrates.

This report has four aims: first, we explore how minor sequence substitutions in the 29-RKRSR-33 target sequence in human H2B affect PRMT7 methylation activity. Second, we aim to identify key residues and motifs that promote methylation of histone H2B (23–37) by GST-*Hs*PRMT7. Third, we measure the contributions of specific residues and motifs in the 29-RKRSR-33 target sequence with Michaelis-Menten kinetic experiments. Fourth, we use the Michaelis-Menten kinetic model to explore the inhibitory effect of ionic strength on PRMT7 methylation activity. We confirm that even subtle substitutions in the RXR recognition motif in substrate peptides dramatically affect PRMT7 methylation activity. Additionally, we show that the presence of an Arg-X-Arg (RXR) motif is a key substrate feature promoting methylation by PRMT7. Further, we suggest that changes to the 29-RKRSR-33 target sequence lead to lower PRMT7 methylation activity *via* decreased catalytic turnover. Lastly, we show that increased ionic strength can inhibit PRMT7 methylation activity primarily by decreasing apparent enzyme-substrate binding affinity.

## Materials and methods

### Preparation of proteins, peptide substrates, and full-length histone substrates

Human PRMT7 used in this study was expressed in BL21 DE3 *E. coli* as a fusion protein of full-length (residues 1–218) glutathione S-transferase from *Schistosoma japonicum* (GST) and

**Table 1. Summary of protein and peptide substrates.**

| Protein or Peptide Name | Sequence | Theoretical[c,d] m.w. (Da) | Experimental[d] m.w. (Da) | HPLC Purity[e] |
|---|---|---|---|---|
| *H. sapiens* H2B | - [a] | 13,906 [a,c] | — | — |
| *X. laevis* H2B | - [b] | 13,934 [b,c] | — | — |
| *H. sapiens* H2B (23–37) | Ac-KKDGKK**R**K**R**S**R**KESY-amide | 1935.25 | 1935.6 | 87.8% |
| *X. laevis* H2B (23–37) | KKDGKK**RR**KS**R**KESY | 1894.20 | 1894.0 | 88.8% |
| Mono R | Ac-KKGG**R**GGGGKKY-amide | 1234.39 | 1233.0 | 95.5% |
| RGR | Ac-KKGG**R**G**R**GGKKY-amide | 1332.56 | 1332.3 | 92.0% |
| KRGR | Ac-KKGK**R**G**R**GGKKY-amide | 1403.69 | 1403.2 | 96.0% |
| RKR | Ac-KKGG**R**K**R**GGKKY-amide | 1403.69 | 1403.2 | 97.9% |
| RGRK | Ac-KKGG**R**G**R**KGGKKY-amide | 1403.69 | 1403.2 | 98.1% |
| RER | Ac-KKGG**R**E**R**GGKKY-amide | 1404.63 | 1404.4 | 97.0% |

[a] Obtained from UNIPROT accession number P62807.

[b] Obtained from UNIPROT accession number P02281.

[c] Theoretical molecular weight using Expasy Compute pI/Mw tool.

[d] Theoretical/experimental molecular weight determined by GenScript.

[e] Experimentally determined HPLC purity reported on GenScript peptide analysis certificate.

full-length human PRMT7 (residues 1–692) linked by the sequence SDLVPRGSST (Addgene. org plasmid no. 34693) [21]. We designated this GST-human PRMT7 fusion protein as GST-*Hs*PRMT7. Transformed cells were lysed, followed by purification of GST-*Hs*PRMT7 *via* column chromatography with glutathione-Sepharose beads as described previously [9]. Recombinant full-length human histone H2B was purchased from New England Biolabs (NEB catalog no. M2505S). Full-length *Xenopus laevis* histone H2B was received as a generous gift from Maria Vogelauer and Siavash Kurdistani at the Department of Biological Chemistry at UCLA [22]. Synthetic peptides used in this study included human histone H2B (23–37), *X. laevis* H2B (23–37), "Mono R," "RGR," "KRGR," "RKR," "RGRK," and "RER" which were purchased from GenScript. At least 85% HPLC purity and TFA salt removal was guaranteed by GenScript. All peptides arrived in lyophilized form and were dissolved in distilled water to form 10 mM stock solutions and were further diluted in water to form 100 μM and 500 μM stock solutions to be used in methylation assays described in this study. A summary table of all protein and peptide sequences is included in Table 1.

### *In vitro* methylation assay of GST-*Hs*PRMT7 with protein and peptide substrates

*In vitro* methylation assays were performed with GST-*Hs*PRMT7, the indicated substrate protein or peptide, and 1.4 μM *S*-adenosyl-L-[*methyl*-[3]H]methionine ([3]H]-AdoMet; PerkinElmer cat no. NET155V), prepared from diluting a stock of 7.0 μM [3]H]-AdoMet (81.9 Ci/mmol, 1 mCi/mL in 9:1 10 mM $H_2SO_4$:EtOH, v/v) in 10 mM HCl. All reactions were performed using distilled water buffered with 50 mM potassium HEPES, pH 9.5 and 1 mM DTT, reaching a final reaction volume of 30 μL. The reactions were initiated by adding [3]H]-AdoMet to a final concentration of 0.14 μM. Control reactions (enzyme-only, substrate-only) contained distilled water instead of substrate or GST-*Hs*PRMT7, respectively. After initiation with [3]H]-AdoMet, all reactions analyzed *via* P81 phosphocellulose filter paper assay were incubated in a water bath at 25°C unless otherwise indicated for the specified time periods and were terminated with 0.5 μL 100% (w/v) trifluoroacetic acid (TFA). P81 assays performed at t = 0 were

individually terminated immediately following initiation. Samples analyzed *via* SDS-PAGE were incubated at 4˚C for 24 hours and were quenched by adding 5x SDS sample loading buffer (250 mM Tris-HCl, pH 6.8; 10% SDS, 30% glycerol, 10% β-mercaptoethanol, 0.2% bromophenol blue), followed by heating at 97˚C for 3 min.

## SDS-PAGE and fluorography of methylated full-length histones

*In vitro* methylation assays with GST-*Hs*PRMT7, full-length histone H2B substrates, and [3H]-AdoMet were set up as described previously [9]. Reaction samples with human and *X. laevis* histone H2B contained 1 μg and 3 μg, respectively. After quenching, samples were immediately loaded onto a 4–20% Bis-tris 10-well ExpressPlus PAGE gel (GenScript catalog no. M42010). Gel electrophoresis was performed for approximately 60 min at 140 V and 400 mA (BioRad PowerPac 300). The gel was stained with Coomassie Blue for one hour followed by overnight destain in 15% methanol and 10% acetic acid. After soaking the destained gel overnight in water, the gel was treated with EN3HANCE autoradiography fluid as per the manufacturer's recommendations (PerkinElmer catalog no. 6NE9701) and dried under vacuum. To visualize species methylated with [3H]-methyl groups, the gel was exposed to autoradiography film at -80˚C for 1 and 74 days, and then [3H] fluorographs were developed and digitally scanned.

## P81 phosphocellulose filter paper methylation assay

*In vitro* methylation assays with GST-*Hs*PRMT7, selected methyl-accepting substrates, and [3H]-AdoMet were set up as described previously [9] and performed in triplicate. Following incubation at 25˚C unless otherwise indicated for the time periods specified, all samples were quenched with 0.5 μL 100% TFA and centrifuged at 9,300 *g* for 10 seconds in an Eppendorf centrifuge (Model 5417C). 1 cm$^2$ pieces of P81 phosphocellulose filter paper (LabAlley catalog no. 05-717-2A) were spotted with 25 μL from each reaction mixture and set to dry overnight. The following day, all filter paper pieces were washed three times with 50 mM sodium bicarbonate solution, pH 9.0, and dried again in liquid scintillation vials. After 5 mL of Safety Solve fluor (RPI industries) was added to each vial, [3H] radioactivity in each sample was quantified *via* liquid scintillation counter (Beckman LS6500) for four total cycles at 5 minutes per sample. All data presented are expressed as averages of the cpm from cycles 2–4.

## Statistical methods

Statistical analyses were performed using the "Analyze Data" function on GraphPad Prism v.8.0.1. To determine statistical significance, unpaired homoscedastic two-tailed Student's *t* tests were performed with significance level α = 0.05 with the relevant data. To obtain apparent $k_m$ and $V_{max}$ values, kinetic data were analyzed by non-linear regression using the Michaelis-Menten equation.

## Results

### GST-*Hs*PRMT7 methylates full-length *Homo sapiens* histone H2B but not full-length *Xenopus laevis* histone H2B

Previous studies have showed that human histone H2B is specifically methylated by PRMT7 at arginine residues 29, 31, and 33 [8]. This region includes Arg-X-Arg (RXR) sequences shown to be specific for PRMT7 methylation [8, 9, 11]. To study the importance of the RXR target sequence for PRMT7 catalysis, we first compared PRMT7 *in vitro* methylation activity with human histone H2B, containing the sequence 29-RKRSR-33, to that with histone H2B from *X. laevis*, which contains instead the sequence 29-RRKSR-33. The sequences adjacent to this

```
H.sapiens    MPDPAKSAPAPKKGSKKAVTKAQKKDGKKRKRSRKESYSVYVYKVLKQVHPDTGISSKAM    60
X.laevis     MPEPAKSAPAPKKGSKKAVTKTQKKDGKKRRKSRKESYAIYVYKVLKQVHPDTGISSKAM    60
             **:*****************:*******:.*****.:.***************

H.sapiens    GIMNSFVNDIFERIAGEASRLAHYNKRSTITSREIQTAVRLLLPGELAKHAVSEGTKAVT   120
X.laevis     SIMNSFVNDVFERIAGEASRLAHYNKRSTITSREIQTAVRLLLPGELAKHAVSEGTKAVT   120
             .********:**************************************************

H.sapiens    KYTSSK    126
X.laevis     KYTSAK    126
             ****:*
```

**Fig 1. H. sapiens histone H2B and X. laevis histone H2B are highly similar.** The amino acid sequences of *H. sapiens* histone H2B (UniProt accession **Q93079**) and *X. laevis* histone H2B (UniProt accession **P02281**) were aligned using Clustal Omega. Highlighted is the region spanning amino acids 23–37 which contains the target sequence 29-RKRSR-33 present in human histone H2B that is methylated by PRMT7. Other than the substitutions at Lys-30 and Arg-31, the sequences of *H. sapiens* and *X. laevis* histone H2B differ at only seven locations.

region are nearly identical in the human and Xenopus proteins (Fig 1). Methylation assays were conducted by incubating bacterially expressed human GST-PRMT7 fusion protein, [³H]-AdoMet, and purified recombinant histone H2B from human and *X. laevis*, respectively, for 24 h as described in **Materials and Methods**. Reaction samples were analyzed by SDS-PAGE. After Coomassie Blue staining, we observed major bands at ~15, 26, and 100 kDa, which corresponded to the expected sizes for human and *X. laevis* H2B, GST, and GST-*Hs*PRMT7, respectively (Fig 2). As controls, we analyzed reaction mixtures containing only PRMT7, human H2B, or *X. laevis* H2B in parallel.

To visualize methylation of H2B substrates by PRMT7, we obtained 1-day and 74-day [³H]-fluorograph exposures of all samples (Fig 2). Only human histone H2B appeared to be methylated after a 1-day exposure. In either the 1-day or the 74-day exposure, no methylation of *X. laevis* H2B was detectable, suggesting that even the minor substitution of Arg-31 and Lys-30 in the 29-RKRSR-33 human H2B sequence leads to loss of RXR sites and elimination of methyl accepting activity for PRMT7.

## The 29-RKRSR-33 sequence in human H2B is critical for GST-*Hs* PRMT7 methylation activity

As a next step, we wanted to investigate whether changes to the 29-RKRSR-33 sequence could explain the loss of human PRMT7 methylation activity with Xenopus H2B. To isolate any possible effect from this target sequence, we designed two synthetic peptides consisting of amino acids 23–37 of histone H2B from human and Xenopus whose sequences are identical except at amino acids positions 30 and 31. Whereas 29-RKRSR-33 is present in human H2B (23–37), Xenopus H2B (23–37) instead contains 29-RRKSR-33, with all other residues being identical. In Fig 3, we assayed Xenopus histone H2B (23–37) peptide with GST-*Hs*PRMT7 for 40 min at 25°C alongside the human histone H2B (23–37) peptide. We show that GST-*Hs*PRMT7 activity is highest with human histone H2B (23–37), whereas no significant methylation activity over background was detected for Xenopus histone H2B (23–37). These results clearly show that even minor substitutions to the 29-RKRSR-33 human H2B sequence can lead to a near total reduction in GST-*Hs*PRMT7 activity, supporting previous findings that this region is a critical target sequence for human PRMT7 [8]. This result confirms that the amino acid changes in this region and not elsewhere is responsible for the loss of PRMT7 activity.

To explore whether reduced human PRMT7 methylation activity with Xenopus histone H2B (23–37) can be explained by decreased enzyme-substrate binding affinity, we raised the

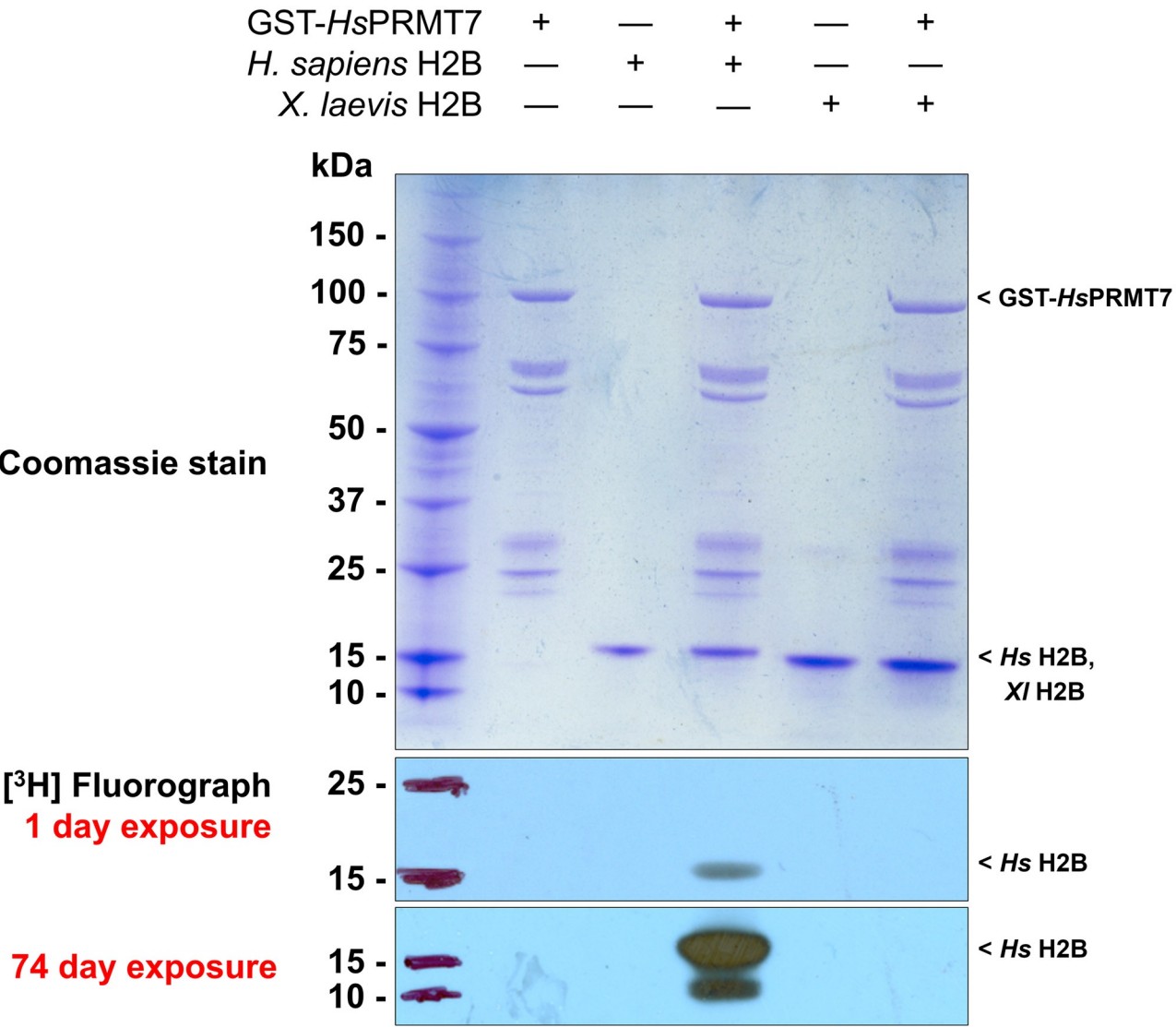

**Fig 2. Xenopus histone H2B is not methylated by human PRMT7.** After incubation for 24 h at 4˚C, reaction samples were terminated and then separated by SDS-PAGE and visualized by Coomassie blue staining (top) and [³H]-fluorography (bottom) as described in Materials and Methods. Samples were exposed for 1 and 74 days. Lane 1 contains protein MW standards. Lanes 2, 3, and 5 are used as negative control and contain GST-*Hs*PRMT7 only, human histone H2B only, and *X. laevis* H2B only, respectively. Lane 4 contains GST-*Hs*PRMT7 and human histone H2B and is used as positive control. Lane 6 contains GST-*Hs*PRMT7 and *X. laevis* histone H2B. Key species present are annotated at their expected MW.

peptide concentration from 10 μM to 50 μM in the experiment performed in Fig 4. Over a time course spanning 0, 1, 2, and 3 hours, we now detected significant PRMT7 methylation activity over background for all peptide substrates that increased linearly over 0–1 hour. Unlike the results shown in Fig 3, these data suggest that Xenopus histone H2B (23–37) is indeed methylated by human PRMT7, albeit at low levels. We also observed that increasing the concentration of *X. laevis* histone H2B (23–37) did not lead to a significant increase in methylation activity under the same conditions, suggesting that PRMT7 was already saturated at the 10 μM peptide concentration. We observed that the measured methylation activity within reaction assays containing 50 μM *X. laevis* histone H2B (23–37) was consistently less

than those containing 10 μM *X. laevis* histone H2B (23–37) at each of the 1, 2, and 3-hour time points (Fig 4), suggesting some substrate inhibition. Conversely, the positive control assays containing enzyme and human histone H2B (23–37) reached approximately 6,000 cpm after just 1 hour, with maximum activity occurring at t = 3 hours at approximately 8,000 cpm (Fig 4). Taken together, the data suggest that the K30R/R31K substitutions in the 29-RKRSR-33 target sequence affects PRMT7 methylation activity by affecting the turnover rate, more so than the apparent substrate binding affinity.

## The RXR motif is necessary for PRMT7 methylation activity

After observing the significant change in PRMT7 activity with just a minor sequence substitution in the methyl-accepting substrate from 29-RKRSR-33 to 29-RRKSR-33, we aimed to identify the residues and sequence motifs that were necessary for PRMT7 catalysis. Previous studies indicated that one crucial motif for PRMT7 catalysis is Arg-X-Arg (RXR), consisting of two arginine residues separated by any other amino acid [8, 11]. Thus, we designed two simple peptides that contain only Arg and Gly in the target sequence and Lys and Gly in the surrounding contextual region: Ac-KKGG**R**GGGGKKY-amide ("mono R") and Ac-KKGG**R**G**R**GGKKY-amide ("RGR"). A single tyrosine residue was added to the C-terminus to quantify peptide concentration by $A_{280}$ absorption. Since Tyr also exists on the C-terminus of human H2B (23–37), we did not expect any major effects from this inclusion. The sequence, predicted molecular weight, and experimentally determined molecular weight of these peptides are summarized in Table 1.

Initially, we compared the PRMT7 methylation activity with human H2B (23–37), mono R, and RGR peptide substrates at t = 20 hours in Fig 5, similar to the experiments performed in Fig 3. As expected, we observed the highest PRMT7 methylation activity with the human H2B (23–37) peptide substrate (Fig 5). Comparatively, PRMT7 activity with the RGR peptide decreased about 4 to 5-fold, whereas methylation of the mono R peptide was barely detectable over the PRMT7-only control.

As a next step, we sought to model the GST-*Hs*PRMT7 methylation with Michaelis-Menten kinetics in order to determine whether the activity differences with each peptide substrate can be explained by changes in apparent binding affinity ($k_m$) or catalytic turnover ($V_{max}$). We designed a time-course methylation experiment similar to Fig 4 to identify the linear initial rate region and investigate the contribution of the arginine residue(s) within the mono R and RGR peptides toward PRMT7 methylation activity. Fig 6 illustrates the change in methylation activity over time from 0, 15, 30, 60, and 120 minutes for reaction mixtures containing GST-*Hs*PRMT7 and either 10 μM mono R peptide or 10 μM RGR peptide. For comparison, we also included reaction assays containing enzyme and 10 μM Xenopus histone H2B (23–37) (Fig 6). As positive control, we compared all assays to reactions containing enzyme and 10 μM human histone H2B (23–37). As negative control, we included reaction assays containing enzyme only. As expected, reactions containing enzyme and human histone H2B (23–37) peptide showed high methylation activity as early as t = 15 min, reaching a maximum value of about 4,000 cpm at t = 120 min. By comparison, the RGR peptide showed a roughly linear increase in cpm from t = 0 to t = 120 min, reaching a maximum value of about 800 cpm at t = 120 min. *X. laevis* histone H2B (23–37) showed low methylation activity but was significantly above control. Methylation activity with samples containing mono R peptide was not significantly greater than control. These data demonstrate that the presence of the RXR motif in the substrate sequence is important for methylation by human PRMT7. However, the substantial difference in activity between the RGR and human H2B (23–37) peptides suggest that other

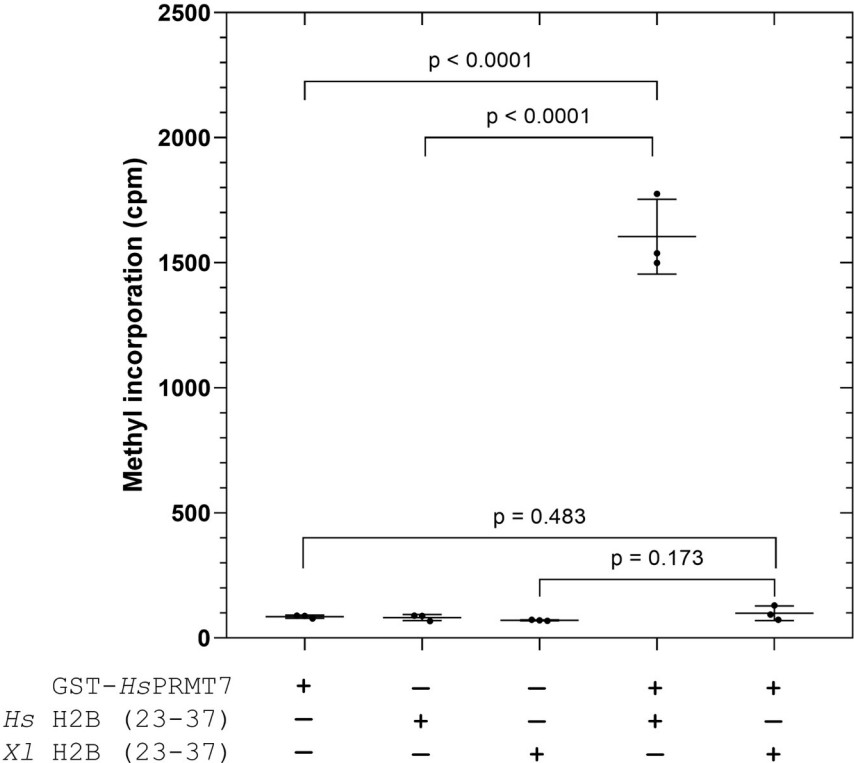

**Fig 3. Methylation of Xenopus histone H2B (23–37) peptide by human PRMT7 is undetectable.** The methylation activity of GST-*Hs*PRMT7 with *H. sapiens* histone H2B (23–37) and *X. laevis* histone H2B (23–37) substrates was assessed using the P81 phosphocellulose paper assay as described in **Materials and Methods**. 5 µg of GST-*Hs*PRMT7 was incubated with 10 µM of the indicated substrate and 0.14 µM [³H]-AdoMet for 40 min at 25˚C. Statistical significance was determined *via* Student's t-test. All reactions indicated were run in triplicate. *Hs* H2B (23–37), *H. sapiens* histone H2B (23–37) peptide; *Xl* H2B (23–37), *X. laevis* histone H2B (23–37) peptide.

residues within the 29-RKRSR-33 sequence also have a role in optimizing enzyme-substrate binding or catalysis.

## Differences in methylation of substrates appears to result from changes in the rate of catalysis and not in the binding affinity

The results in Fig 6 demonstrated a large difference in GST-*Hs*PRMT7 methylation activity between the RGR peptide and the human H2B (23–37) peptide used as positive control despite the presence of the RXR motif in both peptides. As such, we sought to explore whether the presence of charged residues in the target sequence in conjunction with the RXR motif could lead to higher methylation activity. To do this, we first designed four additional synthetic peptides (KRGR, RKR, RGRK, and RER) with nearly identical sequences to the mono R and RGR peptides shown previously, save for an additional lysine or glutamate residue within the target sequence (Table 1). To determine the effect of the RXR motif and charged residues on GST-*Hs*PRMT7 binding and catalysis, we modeled the methylation activity with each peptide substrate with Michaelis-Menten kinetic experiments and compared each peptide's calculated apparent $k_m$ and $V_{max}$ values (Fig 7).

From the experiments shown in Fig 7, we show that the RGR, KRGR, RKR, and RGRK peptides have very similar binding affinities ranging from 0.57 to 0.85 µM whereas the

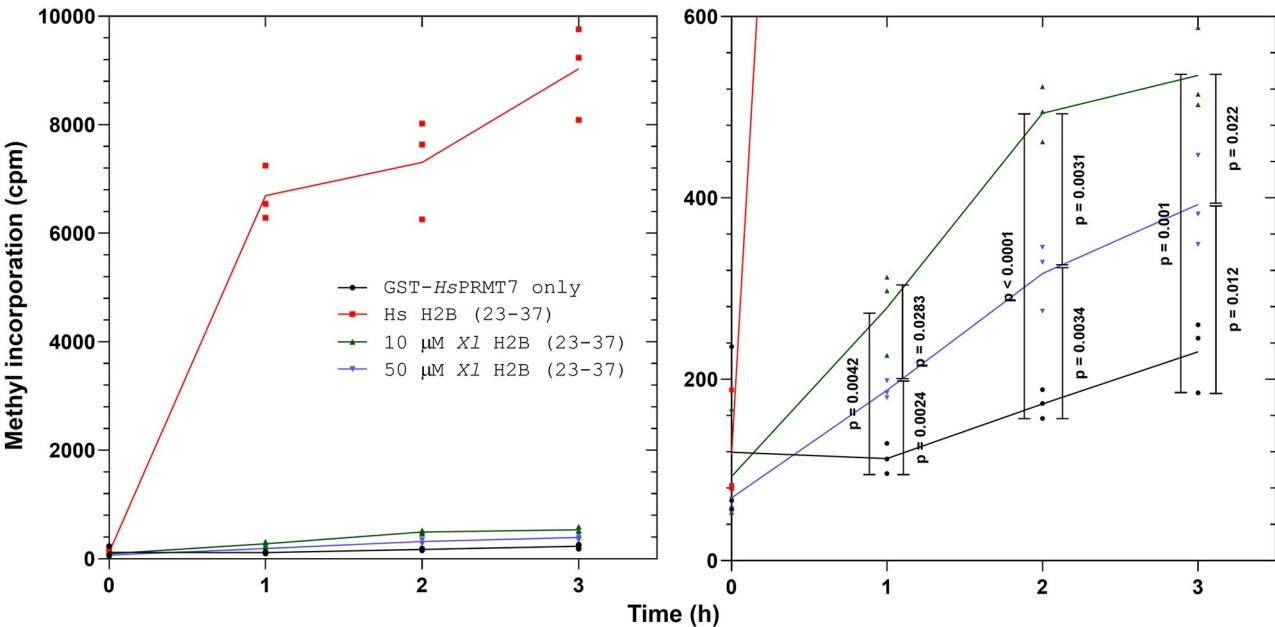

**Fig 4. Comparison of human PRMT7-mediated methylation of human and Xenopus H2B (23–37) peptide substrates over time.** The methylation activity of GST-*Hs*PRMT7 with human histone H2B (23–37) and Xenopus histone H2B (23–37) substrates was assessed using the P81 phosphocellulose paper assay as described in **Materials and Methods**. *Left-hand side*: radioactive counts-per-minute of reactions containing 5 µg of GST-*Hs*PRMT7 incubated with 10 µM *Hs* H2B (23–37), 10 µM *Xl* H2B (23–37), 50 µM *Xl* H2B (23–37), and 0.14 µM [³H]-AdoMet for 0, 1, 2, and 3 h at 25°C. *Right-hand side*: statistical significance was determined for GST-*Hs*PRMT7 only, 10 µM *Xl* H2B (23–37), and 50 µM *Xl* H2B (23–37) at each time point *via* Student's t-test. All reactions indicated were run in triplicate. *Hs* H2B (23–37), *H. sapiens* histone H2B (23–37) peptide; *Xl* H2B (23–37), *X. laevis* histone H2B (23–37) peptide.

corresponding $V_{max}$ values varied significantly from 200 to 1400 µM s$^{-1}$. Comparatively, the mono R and RER peptides showed slightly increased apparent $k_m$ values of 1.4 µM and 2.8 µM, respectively, as well as strongly decreased $V_{max}$ values. Conversely, the calculated $V_{max}$ values varied substantially depending on the presence of the RXR motif and charged residues within the peptide target sequence (Fig 7). We observed the highest $V_{max}$ values with RKR, followed by KRGR, RGR and RGRK. In contrast, the $V_{max}$ values calculated for Mono R and RER were 5–6 times lower than RKR. Taken together, these data suggest that PRMT7 activity could be enhanced by the presence of a positively charged Lys residue within the targeted RXR motif.

## Increases in salt concentration inhibits PRMT7 by decreasing enzyme-substrate binding affinity

Previous studies indicate that GST-*Hs*PRMT7 activity with human H2B (23–37) peptide substrate decreases by half with the addition of 50 mM NaCl or KCl [9]. Comparatively, a previous study of ionic strength in human HEK293 cells estimates the intracellular concentration of potassium ions to be about 120–140 mM in human cells [23, 24]. As such, our results in Fig 7 led us to question whether this dramatic decrease in activity may be associated with a loss of either substrate-enzyme binding affinity or catalytic activity.

 To explore this question, we compared the PRMT7 methylation activity of the human histone H2B (23–37), mono R, RGR, KRGR, RKR, RGRK, and RER peptides *via* P81 methylation assay with and without addition of 130 mM KCl (Fig 8). Apparent $k_m$ and $V_{max}$ values

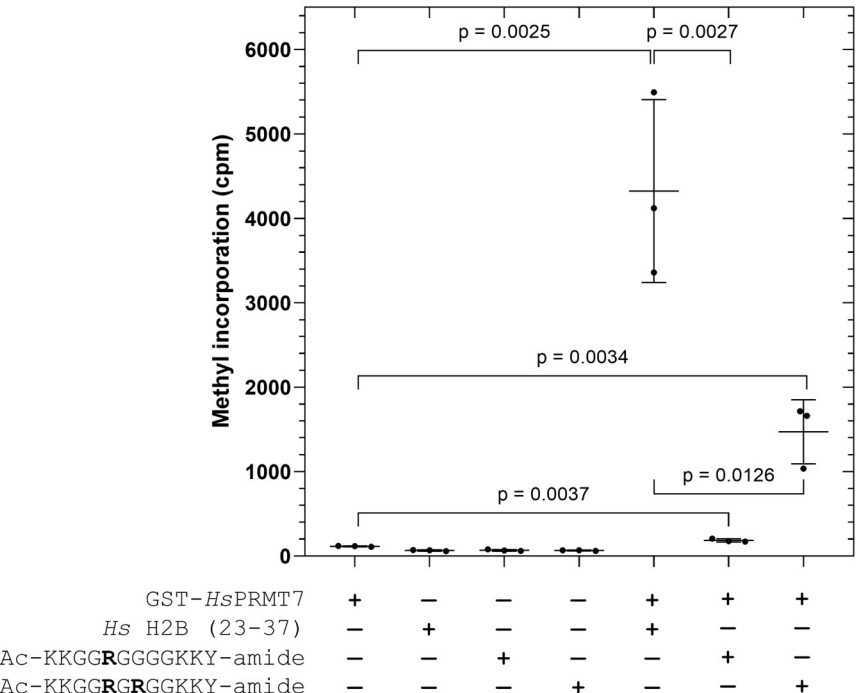

**Fig 5. The effect of the RXR motif on substrate methylation by human PRMT7.** The methylation activity of GST-*Hs*PRMT7 with *Hs* H2B (23–37), mono R, and RGR peptide substrates was assessed using the P81 phosphocellulose paper assay as described in **Materials and Methods**. 5 μg of GST-*Hs*PRMT7 was incubated with 10 μM of the indicated substrate and 0.14 μM [$^3$H]-AdoMet for 20 h at 4˚C. Statistical significance was determined *via* Student's t-test. All reactions indicated were run in triplicate. *Hs* H2B (23–37), *H. sapiens* histone H2B (23–37) peptide; mono R, Ac-KKGG**R**GGGGKKY-amide; RGR, Ac-KKGG**R**G**R**GGKKY-amide.

calculated for each peptide under these conditions are summarized in Table 2. As shown in Fig 8, adding 130 mM KCl to reaction assays showed significantly increased apparent $k_m$ values for all indicated peptide substrates. This apparent $k_m$ increase was up to 30-fold, in the case of the human histone H2B (23–37) control peptide. Moreover, the mono R, KRGR, RKR, RGRK, and RER peptides showed roughly linear initial PRMT7 methylation activity in the chosen concentration range with 130 mM KCl present, further suggesting that the decrease in methylation activity occurs *via* decreases in enzyme-substrate binding affinity. However, it is unclear whether the inhibitory effect of ionic strength in these experiments is purely competitive. For instance, whereas the $V_{max}$ values with and without added salt remained relatively unchanged for human histone H2B (23–37), this was not the case with the RGR, KRGR, RKR, and RGRK peptides, which showed up to an approximately three-fold decrease in $V_{max}$ after adding 130 mM KCl. Taken together, these data suggest that ionic strength primarily affects human PRMT7 substrate binding.

## Discussion

We confirm that human PRMT7 activity is quite sensitive to changes in the RXR target sequence in the substrate protein. The observed loss of PRMT7 methylation activity with the K30R/R31K substitutions in the human histone H2B sequence complements previous findings, where either R29K or R33K substitutions in human histone H2B (23–37) peptides led to at least a two-thirds reduction of PRMT7 activity and an R31K-substituted human H2B (23–

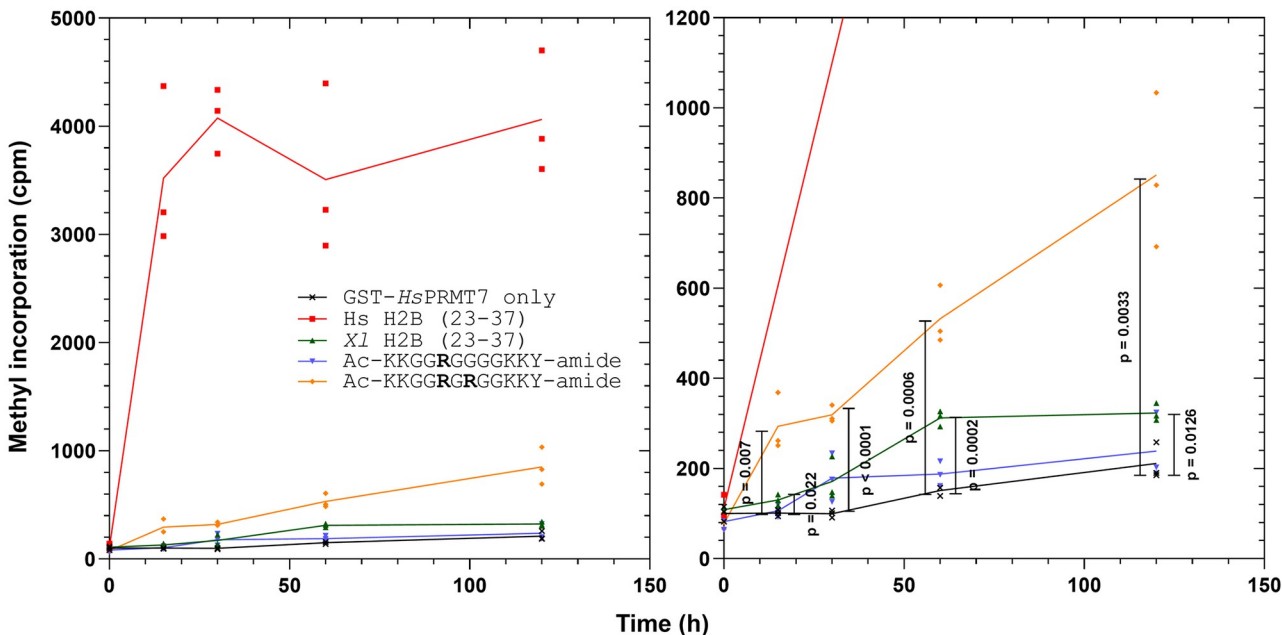

**Fig 6. Comparison of human PRMT7-mediated methylation of human H2B (23–37), Xenopus H2B (23–37), and synthetic peptide substrates over time.** The methylation activity over time of GST-*Hs*PRMT7 with human histone H2B (23–37), Xenopus histone H2B (23–37), mono R and RGR peptide substrates was compared using the P81 phosphocellulose paper assay as described in **Materials and Methods**. *Left-hand side*: radioactive counts-per-minute of reactions containing 5 μg of GST-*Hs*PRMT7 incubated with 10 μM *Hs* H2B, *Xl* H2B (23–37), mono R, or RGR peptide and 0.14 μM [³H]-AdoMet for 0, 15, 30, 60, and 120 min at 25˚C. *Right-hand side*: statistical significance was determined for GST-*Hs*PRMT7 only, *Xl* H2B (23–37), mono R, and RGR at each time point *via* Student's t-test. All reactions indicated were run in triplicate. *Hs* H2B (23–37), *H. sapiens* histone H2B (23–37) peptide; *Xl* H2B (23–37), *X. laevis* histone H2B (23–37) peptide; mono R, Ac-KKGG**R**GGGGKKY-amide; RGR, Ac-KKGG**R**G**R**GGKKY-amide.

37) peptide showed no methyl-accepting activity [11]. The exquisite specificity of PRMT7 toward the basic RXR motif can be contrasted to the much broader specificity of the major type I protein arginine methyltransferase (PRMT1) and the major type II enzyme (PRMT5). These latter enzymes have been shown to methylate multiple amino acid sequences [11, 25]. PRMT1, the major type I PRMT and the enzyme responsible for ~85% of protein arginine methylation, was initially thought to target specific glycine-arginine rich (RGG) motifs such as GG**R**G**R**GG in human fibrillarin and G/FGG**R**GGG/F in heterogenous nuclear ribonuclear proteins [26–28]. However, later work showed that PRMT1 can methylate a much wider range of substrates than that predicted by the RGG consensus sequence, including histone H2AR11 (8-GKA**R**AK-13) and histone H4R3 (1-SG**R**GK-5) [29–33]. PRMT5, the major type II arginine methyltransferase, has also been shown to target a wide range of substrates containing an RG/GRG sequence, although the involvement of substrate-binding adaptor proteins such as pIC1n, RIOK1, and COPR5 in PRMT5-mediated methylation is distinct from the other PRMTs [34]. Methylation studies performed *in vitro* with *Trypanosoma brucei* PRMT7 (*Tb*PRMT7) and peptides based on amino acids 1–21 of histone H4 (1-SG**R**GK-5) showed that *Tb*PRMT7 also targeted GRG substrate motifs, in a distinct specificity than the mammalian isoforms [3, 35, 36]. PRMT9, a minor type II methyltransferase, also appears to be very specific, as to date it has only been shown to methylate the splicing factor SF3B2 at Arg508 (505-CFK**R**KYL-511) with reported $k_m$ value of 0.08 μM [7, 37, 38]. Unlike the other PRMTs, PRMT9 does not appear to methylate either core histones or RGG-containing sequences [7].

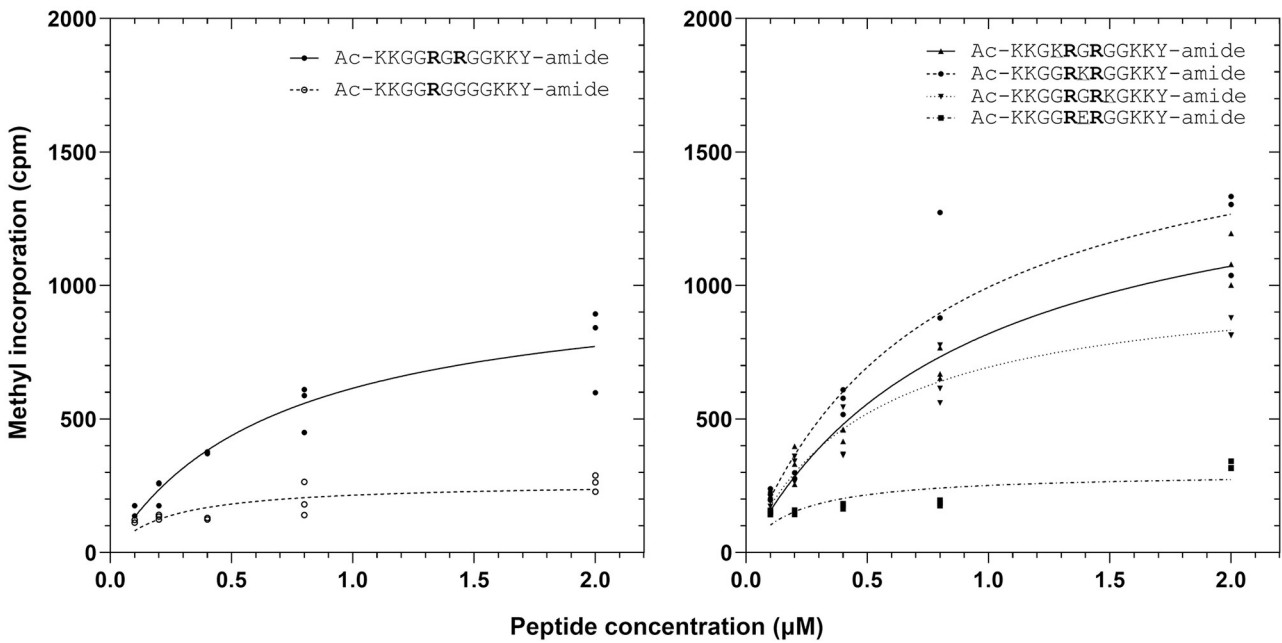

**Fig 7. The effect of the RXR motif and charged residues on PRMT7-substrate enzyme kinetics.** The apparent binding affinity $k_m$ and maximum initial activity $V_{max}$ of GST-*Hs*PRMT7 with synthetic peptide substrates was modeled with Michaelis-Menten kinetics using data gathered by P81 phosphocellulose paper assay as described in **Materials and Methods**. *Left*, methylation of mono R compared to RGR substrates. *Right*, comparison of methylation of KRGR, RKR, RGRK, and RER substrates. 5 μg of GST-*Hs*PRMT7 was incubated with various concentrations of the indicated peptide and 0.14 μM [³H]-AdoMet for 60 min at 25°C. All reactions indicated were run in triplicate. Human histone H2B (23–37) peptide was used as a positive control (not shown). Mono R, Ac-KKGG**R**GGGGKKY-amide; RGR, Ac-KKGG**R**G**R**GGKKY-amide; KRGR, Ac-KKGK**R**G**R**GGKKY-amide; RKR, Ac-KKGG**R**K**R**GGKKY-amide; RGRK, Ac-KKGG**R**G**R**KGKKY-amide; RER, Ac-KKGG**R**E**R**GGKKY-amide.

Interestingly, a recent study presented *in vitro* and *in silico* evidence that human PRMT7 methylates R548 (544-SPC**R**DSV-551) and R753 (750-FCG**R**KQF-756) on the C-terminus of proliferator-activated receptor gamma coactivator-1 alpha [39]. These results could importantly imply the existence of PRMT7 substrates that do not conform to the RXR consensus sequence illustrated here. In a recent review Halabelian and Barsyte-Lovejoy summarized in their Table 1 reports of PRMT7 modification of different proteins [40]. We note that many of these suggested arginine methylation sites have sequences distinct from the Arg-X-Arg motif sequences studied here. Further work will be needed to confirm these distinct sites and to evaluate whether PRMT7 can target distinct types of methyl-accepting sites. We do note that the proposed methylation sites of histone H4 at Arg3, determined indirectly, may in fact reflect methylation by PRMT5 [17]. Additionally, we recognize that there may be significant differences in the recognition of Arg residues by PRMT7 in structured proteins as opposed to unstructured peptides.

At present, it is generally understood that PRMT-substrate recognition and methylation depend on both specific sequence motifs as described above as well as the surrounding primary amino acid context [25]. Previous work suggested that suitable substrates for PRMT7 can contain an RXR motif surrounded by basic residues, as in the lysine- and arginine-rich amino acid 23–37 region of human histone H2B (23-KKDGKK**R**K**R**S**R**KESY-37) [4, 8, 11]. As such, we were interested in isolating critical residues within the 29-RKRSR-33 sequence in human H2B while also minimizing the effect of contextual amino acids. To approach this goal, our synthetic peptide design prioritized the 29-RKRSR-33 target sequence, including only

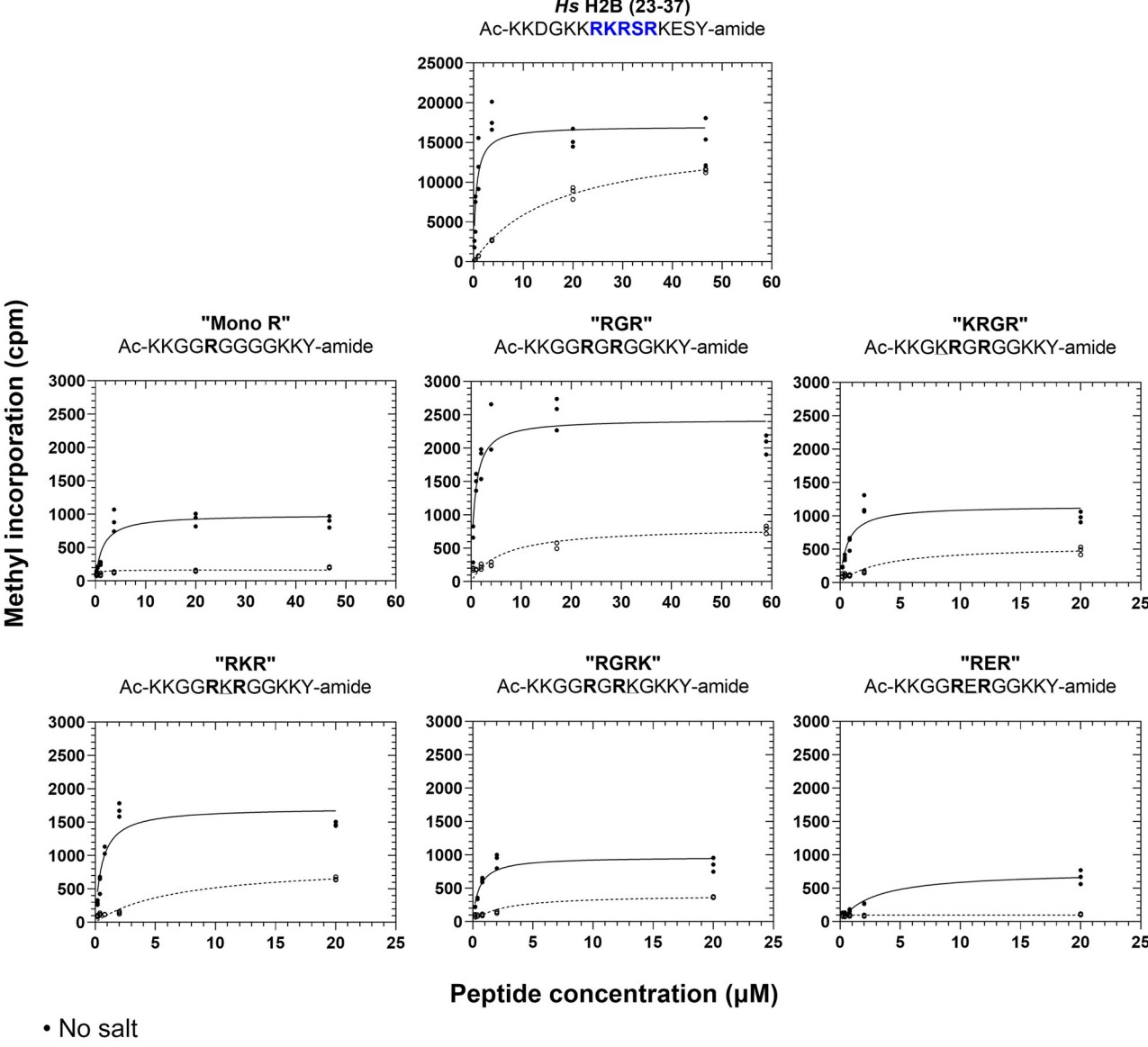

**Fig 8. Increases in ionic strength affects PRMT7-substrate binding affinity and maximum initial reaction velocity.** The apparent binding affinity $k_m$ and maximum initial activity $V_{max}$ of GST-*Hs*PRMT7 with synthetic peptide substrates was modeled with Michaelis-Menten kinetics using data gathered by P81 phosphocellulose paper assay as described in **Materials and Methods**. For all reactions, 5 μg of GST-*Hs*PRMT7 was incubated with various concentrations of the indicated peptide and 0.14 μM [³H]-AdoMet for 60 min at 25˚C. *Closed circle*, methylation activity of human PRMT7 without added ionic strength. *Open circle*, methylation activity of human PRMT7 in the presence of 130 mM KCl. All reactions indicated were run in triplicate. Human histone H2B (23–37) peptide was used as a positive control for all experiments. Mono R, Ac-KKGG**R**GGGGKKY-amide; RGR, Ac-KKGG**R**G**R**GGKKY-amide; KRGR, Ac-KKGK**R**G**R**GGKKY-amide; RKR, Ac-KKGG**R**K**R**GGKKY-amide; RGRK, Ac-KKGG**R**G**R**KGKKY-amide; RER, Ac-KKGG**RER**GGKKY-amide.

lysine and glycine as context. Here, we show that the *in vitro* methylation activity of human PRMT7 decreased dramatically with peptides that either contain a negatively charged Glu within the RXR motif or lack the RXR motif entirely. However, the fact that the highest PRMT7 activity observed with the synthetic peptides is roughly one-fifth of that observed with human histone H2B (23–37) suggests that additional residues in human H2B (23–37)

**Table 2. Summary of calculated Michaelis-Menten kinetic parameters from the data presented in Fig 8.**

| Peptide Name | Peptide Sequence | $k_m$ (µM) | 95% CI | $V_{max}$ (µM s⁻¹) | 95% CI |
|---|---|---|---|---|---|
| **No added salt** | | | | | |
| *Hs* H2B (23–37) | Ac-KKDGKK**R**K**R**S**R**KESY-amide | 0.6 | [0.3, 1.0] | 17000 | [15000, 19000] |
| Mono R | Ac-KKGG**R**GGGGKKY-amide | 1.4 | [0.9, 2.3] | 990 | [870, 1100] |
| RGR | Ac-KKGG**R**G**R**GGKKY-amide | 0.7 | [0.4, 1.2] | 2400 | [2200, 2700] |
| KRGR | Ac-KKGK**R**G**R**GGKKY-amide | 0.6 | [0.3, 1.0] | 1200 | [950, 1400] |
| RKR | Ac-KKGG**R**K**R**GGKKY-amide | 0.6 | [0.3, 0.9] | 1700 | [1500, 2000] |
| RGRK | Ac-KKGG**R**G**R**KGKKY-amide | 0.5 | [0.3, 0.8] | 970 | [840, 1100] |
| RER | Ac-KKGG**R**E**R**GGKKY-amide | 2.8 | [1.7, 4.6] | 750 | [640, 880] |
| **+130 mM KCl** | | | | | |
| *Hs* H2B (23–37) | Ac-KKDGKK**R**K**R**S**R**KESY-amide | 17 | [14, 20] | 16000 | [15000, 17000] |
| Mono R | Ac-KKGG**R**GGGGKKY-amide | —* | —* | —* | —* |
| RGR | Ac-KKGG**R**G**R**GGKKY-amide | 6.2 | [3.5, 11] | 820 | [690, 980] |
| KRGR | Ac-KKGK**R**G**R**GGKKY-amide | 3.3 | [1.7, 7.0] | 550 | [440, 700] |
| RKR | Ac-KKGG**R**K**R**GGKKY-amide | 6.9 | [3.8, 16] | 880 | [710, 1200] |
| RGRK | Ac-KKGG**R**G**R**KGKKY-amide | 2.4 | [1.3, 4.5] | 400 | [330, 490] |
| RER | Ac-KKGG**R**E**R**GGKKY-amide | —* | —* | —* | —* |

All kinetic values and their 95% confidence intervals were obtained using the "statistical analysis" function in GraphPad Prism version 8.0.1.

*indicates poor fit of Michaelis-Menten equation

contribute to PRMT7 catalysis, or that there are additional three-dimensional contacts with residues outside of the motif. Previous mass spectrometry data indicate that PRMT7 methylates human H2B (23–37) at R29, R31, and R33, so it is possible that the third Arg present in human H2B (23–37) also contributes substantially to PRMT7 methylation [8].

We wanted lastly to investigate how the presence of ionic strength inhibited PRMT7 activity. We saw this as a useful next step in identifying possible *in vivo* substrates of human PRMT7, which is challenging in part because PRMT7 lacks significant activity at physiological temperature, pH, and ionic strength [9]. Indeed, observed human PRMT7 activity with human H2B (23–37) peptide substrate decreases by half with the addition of 50 mM NaCl or KCl [9]. By comparison, the intracellular concentration of K⁺ ions in human HEK293 cells is estimated to be between 120–140 mM [23, 24]. We showed that the addition of 130 mM KCl primarily decreased apparent PRMT7-peptide binding affinity for the human H2B (23–37) peptide, whereas the maximum velocity seemed relatively stable. However, we were intrigued to find a decrease of both apparent binding affinity and maximum velocity in the case of other peptides. This result may suggest that the inhibitory effect of ionic strength on PRMT7 activity is not wholly competitive in nature.

Recently, it has been demonstrated that increases in ionic strength also inhibits the *in vitro* activity of PRMT1 and PRMT5 [9]. However, one earlier study presented evidence that high salt conditions instead increased PRMT1 expression and ADMA formation in the human endothelial cell line EA.hy926, resulting in upregulation of the Ras homolog gene family member A/Rho-associated protein kinase pathway and inhibition of nitric oxide synthetase activity [41]. Extensions of this work in rat-based models further showed that increased ADMA concentration resulted in decreased nitric oxide synthesis, implying that salt-dependent upregulation of PRMT1 expression is ultimately involved in the onset and development of salt-sensitive hypertension [42, 43]. As such, the physiological significance of the effect of ionic strength on PRMT activity remains unclear.

Taken together, the results of this study offer additional insight into PRMT7 specificity toward the 29-RKRSR-33 sequence in human histone H2B and, more generally, the effect of disrupting the RXR motif on PRMT7 methylation activity. However, more evidence is needed to elucidate additional substrate factors that contribute to PRMT7 catalysis. In this case, structural evidence could be helpful in determining the origin of the specificity of human PRMT7 toward human H2B. Ultimately, the goal of these studies would be to identify criteria that could accurately predict whether a protein of interest could be a substrate of PRMT7. Such criteria could prove useful for identifying protein substrates of PRMT7 *in vivo* and for further elucidating its role in our physiology.

## Supporting information

**S1 File.**
(XLSX)

**S2 File.**
(XLSX)

**S1 Raw images.**
(PDF)

## Author Contributions

**Conceptualization:** Timothy J. Bondoc, Troy L. Lowe, Steven G. Clarke.

**Data curation:** Timothy J. Bondoc.

**Formal analysis:** Timothy J. Bondoc.

**Funding acquisition:** Steven G. Clarke.

**Investigation:** Timothy J. Bondoc.

**Methodology:** Timothy J. Bondoc, Troy L. Lowe, Steven G. Clarke.

**Resources:** Troy L. Lowe, Steven G. Clarke.

**Supervision:** Troy L. Lowe, Steven G. Clarke.

**Writing – original draft:** Timothy J. Bondoc.

**Writing – review & editing:** Timothy J. Bondoc, Troy L. Lowe, Steven G. Clarke.

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
