## [Decision Letter · Decision Letter 0]

21 Mar 2023

PONE-D-23-04780The exquisite specificity of human protein arginine methyltransferase 7 (PRMT7) toward Arg-X-Arg sitesPLOS ONE

Dear Dr. Clarke,

Thank you for submitting your manuscript to PLOS ONE. After careful consideration, we feel that it has merit but does not fully meet PLOS ONE’s publication criteria as it currently stands. Therefore, we invite you to submit a revised version of the manuscript that addresses the points raised during the review process.

 Please submit your revised manuscript  by 20 April. If you will need more time than this to complete your revisions, please reply to this message or contact the journal office at plosone@plos.org. Please include the following items when submitting your revised manuscript:A rebuttal letter that responds to each point raised by the academic editor and reviewer(s). You should upload this letter as a separate file labeled 'Response to Reviewers'.A marked-up copy of your manuscript that highlights changes made to the original version. You should upload this as a separate file labeled 'Revised Manuscript with Track Changes'.An unmarked version of your revised paper without tracked changes. You should upload this as a separate file labeled 'Manuscript'.

We look forward to receiving your revised manuscript.

Kind regards,

A Ganesan

Academic Editor

PLOS ONE

Journal Requirements:

Reviewers' comments:

Reviewer's Responses to Questions

**Comments to the Author**

1. Is the manuscript technically sound, and do the data support the conclusions?

Reviewer #1: Partly

Reviewer #2: Yes

2. Has the statistical analysis been performed appropriately and rigorously? 

Reviewer #1: No

Reviewer #2: Yes

3. Have the authors made all data underlying the findings in their manuscript fully available?

Reviewer #1: Yes

Reviewer #2: Yes

4. Is the manuscript presented in an intelligible fashion and written in standard English?

Reviewer #1: Yes

Reviewer #2: Yes

5. Review Comments to the Author

Reviewer #1: This manuscript explores the substrate specificity of recombinant full length PRMT7 produced from E. coli using recombinant full length human and X.laevis histones and synthetic short peptides. Enzyme assays are carried out using [3H]-AdoMet assays, using SDS-PAGE/fluorographs for full length histones and P81 phosphocellulose filter paper assay for the peptides. The authors conclude that the Arg-X-Arg consensus is needed for the substrates to be methylated efficiently, where X = Gly or Lys, but not Glu. The surrounding sequences around the Arg-X-Arg motif is important, where Lys before or after the retains the Km indicating those to likely interact with PRMT7. Increase in salt concentration affects both Km and Vmax.

While the study support and add to the insight into PRMT7 activity, it is concise and limited to the histone H2B and a set of KG peptides.

The current study is not sufficiently extensive enough to be able to claim that PRMT7 has ‘exquisite’ specificity for Arg-X-Arg sequence as noted in the title. Firstly, because the sequence variation tested is very limited, and the claim can only be made with respect to mono verses di arginine sequences (mono- and di-arginine containing KG peptides). Secondly, of the KG peptide, the RER peptide, which has the Arg-Glu-Arg motif, is just as poor a substrate as the mono R peptide. Thus, it is not all Arg-X-Arg motif that are substrates. Thirdly, the spacing between the two arginine residues have not been tested thoroughly, and only Arg-Arg and Arg-Lys-Ser-Arg sequences are tested within the X.laevis histone / peptide context (H. sapiens H2B vs X. laevis H2B peptide/histones, where there are three arginines in both but with different spacings).

It appears that RXRXR is the most favoured (from this publication, as well as author’s previous JBC 2013 288(52) 37019-37025)? The authors do not discuss how the knowledge gained in this paper may help with understanding / confirming the PRMT7 cellular substrates (those identified to date in cells e.g. Life 2021 11(8)768?). Also it is important to emphasise the limitations of using peptides.

Specific points:

- Likely due to the formatting but the Figure legends and main text difficult to distinguish.

- Fig 2: Labelling of lane is incorrect? The figures do not support the conclusion in text “Line 194-194: “To visualize methylation of H2B substrates by PRMT7, we obtained 1-day and 74-day [3H]-fluorograph exposures of all samples (Figure 2). Only human histone H2B appeared to be methylated after a 1-day exposure.” Should sample lane 3 be GST-HsPRMT7 + Hs H2B? The 4th lane is appears to be wrong too? Should lane 3 and lane 4 labelling be swapped around? This figure is very confusing. Also any reason that 74 day exposure is chosen?

- line 240: “Taken together, the data suggest that the K30R/R31K substitutions in the 29-RKRSR-33 target sequence affects PRMT7 methylation activity by affecting the turnover rate kcat, more so than the substrate binding affinity, km.” ) In Figure 4, only two peptide concentrations tested – no detail kinetic data provided, so cannot discuss about kcat, km based on this data

- Line 242 - Please note that Km is not binding affinity.

- Fig 4/ Fig 6 – Figures need labelling (LHS, RHS) and noted in the legend.

- Figure 6. Difficult to distinguish between the GST-HsPRMT7 only and Ac-KKGGRGRGGKKY-amide peptides. They are both black circle;

- What is the Km of SAM for PRMT7? Is co-substrate saturating conditions at 0.14uM SAM that the assays are being carried out? Also difficult to check the activity as the y-axis are given in cpm. When Michaelis Menten plots are provided for Hs H2B(23-37) (Fig8), are the initial rates calculated for different substrate concentrations at linear range? The activity on all the other figures under these conditions appears not to be in the linear range.

- The KG peptide nomenclature is confusing – in particular when they are noted in a sentence e.g. “Line 342 - From the experiments shown in Figure 7, we show that the RGR KG, KRGR KG, RKR KG, and RGRK KG peptides have very similar binding affinities ranging from 0.57 to 0.85 μM”. Would help the readers if different names than KG are used so there is no mistaking that the “KG” is not part of the sequence.

- Are the plots for Fig7 the same as for Fig8 but with narrower concentration range?

- Table 2. Please provide errors for the Km and Vmax values. Please add nomenclatures for the peptides.

- While the statistical analysis has been carried out, it is difficult to see if the data analysis is carried out for three technical replicates for a single experiment, or three independent experiments. If the former, then more robust data is needed to support the work.

Reviewer #2: A very detailed analysis of the substrate specificity of PRMT7

Page 14 Paragraph starting line 226 is confusing because the increase in activity observed is due to the increased length of time of the experiment (40 mins vs 3 h). However, the author makes it sound like the increase of histone from 10 to 50 microM is the reason. Then in line 232 the author states that decreased methylation was observed with 50 microM histone compared to 10 microM. As measurements were only taken every hour, then the activity could be considered increasing linearly over 0-3 h (not 0-1).

6. PLOS authors have the option to publish the peer review history of their article (what does this mean?). If published, this will include your full peer review and any attached files.

Reviewer #1: No

Reviewer #2: No

---

## [Author Response · Author response to Decision Letter 0]

6 Apr 2023

Reviewer 1,

We wholeheartedly thank you for your time and your helpful suggestions that you have provided. After very careful consideration, we have added information and made changes to the manuscript with your insight in hopes of more accurately and precisely conveying our recent work on PRMT7 activity. 

We address your suggestions and concerns below in the order that they appeared in the original review. 

This manuscript explores the substrate specificity of recombinant full length PRMT7 produced from E. coli using recombinant full length human and X.laevis histones and synthetic short peptides. Enzyme assays are carried out using [3H]-AdoMet assays, using SDS-PAGE/fluorographs for full length histones and P81 phosphocellulose filter paper assay for the peptides. The authors conclude that the Arg-X-Arg consensus is needed for the substrates to be methylated efficiently, where X = Gly or Lys, but not Glu. The surrounding sequences around the Arg-X-Arg motif is important, where Lys before or after the retains the Km indicating those to likely interact with PRMT7. Increase in salt concentration affects both Km and Vmax.

While the study support and add to the insight into PRMT7 activity, it is concise and limited to the histone H2B and a set of KG peptides.

The current study is not sufficiently extensive enough to be able to claim that PRMT7 has ‘exquisite’ specificity for Arg-X-Arg sequence as noted in the title. Firstly, because the sequence variation tested is very limited, and the claim can only be made with respect to mono verses di arginine sequences (mono- and di-arginine containing KG peptides). Secondly, of the KG peptide, the RER peptide, which has the Arg-Glu-Arg motif, is just as poor a substrate as the mono R peptide. Thus, it is not all Arg-X-Arg motif that are substrates. Thirdly, the spacing between the two arginine residues have not been tested thoroughly, and only Arg-Arg and Arg-Lys-Ser-Arg sequences are tested within the X.laevis histone / peptide context (H. sapiens H2B vs X. laevis H2B peptide/histones, where there are three arginines in both but with different spacings).

>We thank the reviewer for these insights. We have now revised the manuscript to point out the limitations of our study. As the reviewer mentioned, we have focused our work on peptides containing the Arg-X-Arg motif. We believe that the large difference in the recognition of the highly similar Xenopus vs. human histone H2B sequence does, in fact, reflect an “exquisite” sensitivity to a single switch of adjacent Lys and Arg residues (see Fig. 2). This of course does not rule out the recognition of PRMT7 of other peptide or protein sequence, but we wanted to highlight the sensitivity that we observed. 

>We should point out that the apparent km for the RGR peptide is about half of that of the RGG peptide and its Vmax is about 2.5-fold higher (Table 2).

>We have now added discussion of Table 1 in the Halabelian and Barsyte-Lovejoy (2021) review mentioned by the reviewer to emphasize that there may be additional motifs for PRMT7 (see below).

It appears that RXRXR is the most favoured (from this publication, as well as author’s previous JBC 2013 288(52) 37019-37025)? The authors do not discuss how the knowledge gained in this paper may help with understanding / confirming the PRMT7 cellular substrates (those identified to date in cells e.g. Life 2021 11(8)768?). Also it is important to emphasise the limitations of using peptides.

>We thank the reviewer for raising this point. As described above, we have now added discussion of the substrates identified in the Life 2021 report. We have also discussed potential limitations of studies with peptide vs. protein substrates. We should point out that of the proteins listed in Table 1 of Life 2021 report referenced above, only the site for EIF1S1 (EIF1-alpha) was confirmed by direct methylation by PRMT7 and this site was indeed an RXR site (53-RRRIR-57). We should also point out that PRMT7 activity can, in some instances, regulate PRMT5 activity (ref. 17). Thus, the studies with the over- or under-expression of PRMT7 in intact cells may affect PRMT5 sites as well as PRMT7 sites. Finally, some of the sites in Table 1 of the Halabelian and Barsyte-Lovejoy review were from papers using FLAG-tagged PRMT7. It is well known that antibodies to the FLAG tag can also pull down PRMT5, resulting in uncertainty in whether a site is methylated by PRMT7 or PRMT5 (refs. 4-5).

Likely due to the formatting but the Figure legends and main text difficult to distinguish.

>We agree with the reviewer that it is difficult to distinguish the figure legends from the main text. We did format the Figure legends and main text as described by the PLoS ONE Submission Guidelines. However, to clarify, we have now added a sentence denoting the end of the figure legends. 

Fig 2: Labelling of lane is incorrect? The figures do not support the conclusion in text “Line 194-194: “To visualize methylation of H2B substrates by PRMT7, we obtained 1-day and 74-day [3H]-fluorograph exposures of all samples (Figure 2). Only human histone H2B appeared to be methylated after a 1-day exposure.” Should sample lane 3 be GST-HsPRMT7 + Hs H2B? The 4th lane is appears to be wrong too? Should lane 3 and lane 4 labelling be swapped around? This figure is very confusing. Also any reason that 74 day exposure is chosen? 

>We thank the reviewer for identifying this unfortunate mistake. The reviewer is correct that lane 3 and lane 4 labeling should be swapped. The Figure 2 legend and main text contain the correct labeling, and lanes in Figure 2 have been re-labeled correctly.

>To address the reasoning behind the 74-day exposure: we intended to expose the “long exposure” film in hopes of detecting any methylation of Xenopus laevis H2B. Ultimately, we developed the film to move forward with the work and recorded 74 days as the total time of exposure.

line 240: “Taken together, the data suggest that the K30R/R31K substitutions in the 29-RKRSR-33 target sequence affects PRMT7 methylation activity by affecting the turnover rate kcat, more so than the substrate binding affinity, km.” ) In Figure 4, only two peptide concentrations tested – no detail kinetic data provided, so cannot discuss about kcat, km based on this data

>We thank the reviewer for pointing out this problem. We have now omitted “kcat” and “km” to indicate that the enzyme was already saturated at the 10 µM concentration.

Line 242- Please note that Km is not binding affinity.

>We thank the reviewer for pointing this out. The text has been changed accordingly.

Fig 4/ Fig 6 – Figures need labelling (LHS, RHS) and noted in the legend.

>We thank the reviewer for this helpful suggestion. The legends of Figures 4 and 6 have been updated accordingly in order to clarify each panel.

Figure 6. Difficult to distinguish between the GST-HsPRMT7 only and Ac-KKGGRGRGGKKY-amide peptides. They are both black circle;

>We thank the reviewer for this important feedback. To make the GST-HsPRMT7 only and Ac-KKGGRGRGGKKY-amide peptide data more easily distinguishable, we have updated Fig 6 with black cross symbols denoting GST-HsPRMT7 only and orange diamond symbols denoting Ac-KKGGRGRGGKKY-amide.

What is the Km of SAM for PRMT7? Is co-substrate saturating conditions at 0.14uM SAM that the assays are being carried out? Also difficult to check the activity as the y-axis are given in cpm. When Michaelis Menten plots are provided for Hs H2B(23-37) (Fig8), are the initial rates calculated for different substrate concentrations at linear range? The activity on all the other figures under these conditions appears not to be in the linear range.

>In this work, we use the same concentration of AdoMet as in the previous paper by Lowe and Clarke (J. Biol. Chem. 2022, or ref. 9 in our manuscript). Since we did not determine the km for AdoMet, we have now revised our paper to report apparent km values for the methyl-accepting peptide in all cases.

In figures 4 and 6, we show time courses indicating linearity for the peptides that were not as active as the human H2B (23-37) peptide. For the most active H2B (23-37) peptide, this might result in an over-estimate of the apparent km.

>Relative activity values are given in cpm.

The KG peptide nomenclature is confusing – in particular when they are noted in a sentence e.g. “Line 342 - From the experiments shown in Figure 7, we show that the RGR KG, KRGR KG, RKR KG, and RGRK KG peptides have very similar binding affinities ranging from 0.57 to 0.85 μM”. Would help the readers if different names than KG are used so there is no mistaking that the “KG” is not part of the sequence.

>We have now modified the peptide nomenclature to use RGR, KRGR, RKR, RGRK, and RER in place of “RGR KG, KRGR KG, RKR KG, RGRK KG” and RER KG.

>Additionally, figure 8 has been updated to reflect these changes.

Are the plots for Fig7 the same as for Fig8 but with narrower concentration range?

>We thank the reviewer for their question. To clarify, the plots for Fig7 and Fig8 are not the same. The data for Fig7 are the apparent Km values obtained under the in vitro conditions we described in Materials and Methods, and the data for Fig8 were obtained with additional follow-up experiments comparing kinetic data with and without the presence of 130mM KCl. We hope this helps address any concerns the reviewer may have.

Table 2. Please provide errors for the Km and Vmax values. Please add nomenclatures for the peptides.

>We thank the reviewer for this helpful suggestion. We have added the 95% confidence interval limits to each entry in Table 2. We have also added the nomenclatures for the peptides as requested.

While the statistical analysis has been carried out, it is difficult to see if the data analysis is carried out for three technical replicates for a single experiment, or three independent experiments. If the former, then more robust data is needed to support the work.

>The methylation at each concentration was determined in triplicate and the apparent km and Vmax values were determined from all of the points leading to the estimated confidence limits.

>We used Prism GraphPad v.8.0.1 to calculate all error values.

 

Reviewer 2,

We wholeheartedly thank you for your time and your encouraging comments that you have provided. After very careful consideration, we have added information and made changes to the manuscript with your insight in hopes of more accurately and precisely conveying our recent work on PRMT7 activity. 

>We address your suggestions and concerns below in the order that they appeared in the original review. 

A very detailed analysis of the substrate specificity of PRMT7

>We thank the reviewer for their positive comment on our work.

Page 14 Paragraph starting line 226 is confusing because the increase in activity observed is due to the increased length of time of the experiment (40 mins vs 3 h). However, the author makes it sound like the increase of histone from 10 to 50 microM is the reason. Then in line 232 the author states that decreased methylation was observed with 50 microM histone compared to 10 microM. As measurements were only taken every hour, then the activity could be considered increasing linearly over 0-3 h (not 0-1).

>Thank you for pointing this out. We have now revised the paragraph to point out that the comparison at 10uM and 50uM was done under the same conditions (Figure 4).

---

## [Decision Letter · Decision Letter 1]

2 May 2023

The exquisite specificity of human protein arginine methyltransferase 7 (PRMT7) toward Arg-X-Arg sites

PONE-D-23-04780R1

Dear Dr. Clarke,

We’re pleased to inform you that your manuscript has been judged scientifically suitable for publication and will be formally accepted for publication once it meets all outstanding technical requirements.

Kind regards,

A Ganesan

Academic Editor

PLOS ONE

Additional Editor Comments (optional):

Reviewers' comments:

Reviewer's Responses to Questions

**Comments to the Author**

1. If the authors have adequately addressed your comments raised in a previous round of review and you feel that this manuscript is now acceptable for publication, you may indicate that here to bypass the “Comments to the Author” section, enter your conflict of interest statement in the “Confidential to Editor” section, and submit your "Accept" recommendation.

Reviewer #1: All comments have been addressed

Reviewer #2: All comments have been addressed

2. Is the manuscript technically sound, and do the data support the conclusions?

Reviewer #1: Yes

Reviewer #2: Yes

3. Has the statistical analysis been performed appropriately and rigorously? 

Reviewer #1: Yes

Reviewer #2: Yes

4. Have the authors made all data underlying the findings in their manuscript fully available?

Reviewer #1: Yes

Reviewer #2: Yes

5. Is the manuscript presented in an intelligible fashion and written in standard English?

Reviewer #1: Yes

Reviewer #2: Yes

6. Review Comments to the Author

Reviewer #1: The authors have addressed all concerns of the reviewers, and the manuscript is clearer to follow for the readers.

Reviewer #2: (No Response)

7. PLOS authors have the option to publish the peer review history of their article (what does this mean?). If published, this will include your full peer review and any attached files.

Reviewer #1: No

Reviewer #2: No

---

## [Editor Report · Acceptance letter]

12 May 2023

PONE-D-23-04780R1 

The exquisite specificity of human protein arginine methyltransferase 7 (PRMT7) toward Arg-X-Arg sites 

Dear Dr. Clarke:

I'm pleased to inform you that your manuscript has been deemed suitable for publication in PLOS ONE. Congratulations! Your manuscript is now with our production department. 

Kind regards, 

on behalf of

Prof. A Ganesan 

Academic Editor

PLOS ONE